# Asbestos Exposure and Malignant Mesothelioma in Construction Workers—Epidemiological Remarks by the Italian National Mesothelioma Registry (ReNaM)

**DOI:** 10.3390/ijerph19010235

**Published:** 2021-12-26

**Authors:** Alessandra Binazzi, Davide Di Marzio, Marina Verardo, Enrica Migliore, Lucia Benfatto, Davide Malacarne, Carolina Mensi, Dario Consonni, Silvia Eccher, Guido Mazzoleni, Vera Comiati, Corrado Negro, Antonio Romanelli, Elisabetta Chellini, Alessia Angelini, Iolanda Grappasonni, Gabriella Madeo, Elisa Romeo, Annamaria Di Giammarco, Francesco Carrozza, Italo F. Angelillo, Domenica Cavone, Luigi Vimercati, Michele Labianca, Federico Tallarigo, Rosario Tumino, Massimo Melis, Michela Bonafede, Alberto Scarselli, Alessandro Marinaccio

**Affiliations:** 1Department of Occupational and Environmental Medicine, Epidemiology and Hygiene, Istituto Nazionale per l’Assicurazione Contro gli Infortuni sul Lavoro, 00143 Roma, Italy; a.binazzi@inail.it (A.B.); d.dimarzio@inail.it (D.D.M.); m.bonafede@inail.it (M.B.); a.scarselli@inail.it (A.S.); 2Valle d’Aosta Health Local Unit, Regional Operating Center of Valle d’Aosta (COR Valle d’Aosta), 11100 Aosta, Italy; mverardo@ausl.vda.it; 3Unit of Cancer Epidemiology, Regional Operating Center of Piemonte (COR Piemonte), University of Torino and CPO-Piemonte, 10124 Torino, Italy; enrica.migliore@cpo.it; 4Regional Operating Center of Liguria (COR Liguria), UO Clinical Epidemiology, IRCCS AOU Policlinico San Martino, 16132 Genova, Italy; lucia.benfatto@hsanmartino.it (L.B.); davide.malacarne@hsanmartino.it (D.M.); 5Epidemiology Unit, Regional Operating Center of Lombardia (COR Lombardia), Fondazione IRCCS Ca’ Granda Ospedale Maggiore Policlinico, 20122 Milano, Italy; carolina.mensi@unimi.it (C.M.); dario.consonni@unimi.it (D.C.); 6Hygiene and Occupational Medicine, Provincial Unit of Health, Regional Operating Center of Autonomous Province of Trento (COR A.P. of Trento), 38100 Trento, Italy; silvia.eccher@apss.tn.it; 7Occupational Medicine Unit, Alto Adige Health Authority, Regional Operating Center of Autonomous Province of Bolzano (COR A.P. of Bolzano), 39100 Bolzano, Italy; guido.mazzoleni@sabes.it; 8Azienda Zero, Epidemiological Department, Regional Operating Center of Veneto (COR Veneto), Veneto Region, 35131 Padova, Italy; vera.comiati@azero.veneto.it; 9Clinical Unit of Occupational Medicine, Regional Operating Center of Friuli-Venezia Giulia (COR Friuli-Venezia Giulia), University of Trieste-Trieste General Hospitals, 34123 Trieste, Italy; negro@units.it; 10Health Local Unit, Public Health Department, Regional Operating Center of Emilia-Romagna (COR Emilia-Romagna), 42020 Reggio Emilia, Italy; romanellia@ausl.re.it; 11Prevention and Clinical Network, Institute for Cancer Research, Regional Operating Center of Toscana (COR Toscana), 50139 Firenze, Italy; e.chellini@ispro.toscana.it (E.C.); a.angelini@ispro.toscana.it (A.A.); 12Regional Operating Center of Marche (COR Marche), School of Medicinal and Health Products Sciences, University of Camerino, 62032 Camerino, Italy; iolanda.grappasonni@unicam.it; 13Regional Operating Center of Umbria (COR Umbria), Servizio Prevenzione, Sanità Veterinaria e Sicurezza Alimentare-Regione Umbria, 06126 Perugia, Italy; gmadeo@regione.umbria.it; 14Regional Operating Center of Lazio (COR Lazio), Department of Epidemiology, Lazio Region, 00143 Roma, Italy; e.romeo@deplazio.it; 15Occupational Medicine Unit, Health Local Unit, Regional Operating Center of Abruzzo (COR Abruzzo), 65121 Pescara, Italy; annamaria.digiammarco@ausl.pe.it or; 16Oncology Unit, Cardarelli Hospital, Regional Operating Center of Molise (COR Molise), 86100 Campobasso, Italy; carrozza_f@yahoo.it; 17Department of Experimental Medicine, “Luigi Vanvitelli” University, Regional Operating Center of Campania (COR Campania), 80138 Napoli, Italy; italof.angelillo@unina2.it or; 18Section of Occupational Medicine ‘‘B.Ramazzini’’, Department of Interdisciplinary Medicine, Regional Operating Center of Puglia (COR Puglia), University of Bari, 70125 Bari, Italy; domenica.cavone@uniba.it (D.C.); luigi.vimercati@uniba.it (L.V.); 19Epidemiologic Regional Center, Regional Operating Center of Basilicata (COR Basilicata), 85100 Potenza, Italy; michele.labianca@supporto.regione.basilicata.it; 20Public Health Unit, Regional Operating Center of Calabria (COR Calabria), 88900 Crotone, Italy; federicotallarigo@libero.it; 21Cancer Registry ASP Ragusa and Sicilia Regional Epidemiological Observatory, Regional Operating Center of Sicilia (COR Sicilia), 97100 Ragusa, Italy; rtuminomail@gmail.com; 22Regional Epidemiological Center, Regional Operating Center of Sardegna (COR Sardegna), 09125 Cagliari, Italy; massimelis@gmail.com

**Keywords:** mesothelioma, asbestos, construction workers, national mesothelioma registry, Italy

## Abstract

Notwithstanding the ban in 1992, asbestos exposure for workers in the construction sector in Italy remains a concern. The purpose of this study is to describe the characteristics of malignant mesothelioma (MM) cases recorded by the Italian registry (ReNaM) among construction workers. Incident mesothelioma cases with a definite asbestos exposure have been analyzed. Characteristics of cases and territorial clusters of crude rates of MM in construction workers have been described, as well as the relation between asbestos use before the ban and the historical trend of workforce in the construction sector in Italy. ReNaM has collected 31,572 incident MM cases in the period from 1993 to 2018 and asbestos exposure has been assessed for 24,864 (78.2%) cases. An occupational exposure has been reported for 17,191 MM cases (69.1% of subjects with a definite asbestos exposure). Among them, 3574 had worked in the construction sector, with an increasing trend from 15.8% in the 1993–98 period to 23.9% in 2014–2018 and a ubiquitous territorial distribution. The large use of asbestos in construction sector before the ban makes probability of exposure for workers a real concern still today, particularly for those working in maintenance and removal of old buildings. There is a clear need to assess, inform, and prevent asbestos exposure in this sector.

## 1. Introduction

Malignant mesothelioma (MM) is a rare cancer with global incidence in the general population estimated at 0.30 cases (per 100,000 inhabitants) in 2020. In Italy, that is among the countries with the highest values (Northern Europe: 1.4; Australia and New Zealand: 1.3) the incidence rate was estimated of one case per 100,000 [1]. MM is associated with occupational, environmental and in house exposures to asbestos fibers and other asbestiform fibers (vermiculite, erionite, fluoro-edenite) [2].

The prognosis remains generally poor with a median survival of approximately 9–12 months [3]. The average latency period, delimited from first causative exposure to malignant mesothelioma diagnosis, is approximately 40 years, with a range from 20 to 70 years [4]. Occupational exposures to asbestos have occurred in a variety of industrial contexts, including asbestos mining and milling, manufacturing of asbestos containing materials (ACMs), shipbuilding and repair, railway carriages maintenance, and construction.

In Italy asbestos was largely used in the past, with 3,748,550 tons of raw asbestos produced up to 1992, and a peak of more than 160,000 tons/year between 1976 and 1980 [5]. Since the early 1980’s, regulations on production and use of asbestos in many developed countries has led to reduction of exposure, particularly in the occupational setting. Italy banned asbestos use in 1992 [6]. The measures included prohibition of extraction, import, export, commercialization and production of asbestos or ACMs. At present, the only permitted activities concern ACMs abatement, disposal and remediation, which attributes the obligation of registration in the national register for companies that perform demolition and/or asbestos removal works [7].

Asbestos has been widely used in building materials and it can still persist in place. In the requirement of applying procedures to remove it, workers employed in abatement and removal of ACMs may be exposed to low but still relevant doses of airborne asbestos fibers [8,9]. Increased risk of MM was reported in construction workers that experienced asbestos exposure in the past [10,11,12,13,14,15,16,17]. Unawareness of indirect exposure may have affected several types of workers, such as carpenters, electricians, plumbers and welders [18,19,20]. Mesothelioma cases occurring during home renovation were also described [21].

European Union directives [22,23] introduced compulsory prevention and protection measures in the workplace, implying that only smaller health effects should be expected from current exposures. However, such effects are not easy to be estimated, due to the long latency of cancer and difficulties in performing job-specific cohort studies. A recent study, based on the Italian surveillance system of occupational exposure to carcinogens (SIREP), evidenced that many construction workers have exposure levels above the action limit established by national legislation (0.01 f/cc), and in a very limited fraction of workers also exceeding the EU limit value (0.1 f/cc) [24].

The aim of this study is to analyze the proportion of constructions workers among all mesothelioma cases registered by the Italian National Registry of Malignant Mesothelioma (ReNaM).

Concern deriving from such exposures and the effectiveness of prevention measures for construction workers dealing with activities in which ACMs are still present will be discussed.

## 2. Materials and Methods

ReNaM is an epidemiological surveillance system characterized by a network of regional operating centers (‘Centri Operativi Regionali’: COR) gradually established in all of the 20 Italian regions. CORs actively search and register incident cases of MM from health care services and obtain occupational and residential history and lifestyle habits interviewing the affected subjects (or next of kin) through a standardized questionnaire. With regard to diagnosis of MM, the cases were allocated to one of three classes that are distinguished by an increasing level of uncertainty: “certain” (with histological confirmation of diagnosis, possibly completed by immune-histochemical characterization, and confirmation by imaging and clinical diagnosis), “probable” (usually, cytological diagnosis and confirmation by imaging and clinical diagnosis) or “possible” MM (clinical diagnosis with positive imaging). Asbestos exposure is categorized as “occupational” (with three degrees of certainty: “definite”, “probable”, “possible”) or “non-occupational” (in-house, environmental, other non-occupational–such as leisure-time-related activities). “Unlikely” exposure is assigned to subjects for whom information is inadequate or asbestos exposure could be reasonably ruled out. More details about the CORs’ procedures and network have been described elsewhere [5,25]. CORs periodically transmit the collected data to the central archive of ReNaM (established at the Italian national workers’ compensation authority (INAIL)) that provides epidemiological nation-wide analyses and promotes specific research projects. Currently, the CORs of Calabria, Sardinia, and Molise cannot warrant the completeness of cases registration, while those of Campania, Abruzzo and Umbria have temporarily suspended their activity due to insufficient resources. To date, ReNaM has collected cases with a diagnosis of MM in the period between 1993 and 2018, whereas the analyses for the period of incidence between 2019 and 2020 are ongoing.

For the aim of this work, from the ReNaM archive we selected cases of MM with an interview, grouped as follows: (1) subjects with an exposure evaluation that includes ‘occupational’, ‘non occupational’ or ‘unlikely’; (2) subjects with an occupational exposure and (3) subjects with an occupational exposure exclusively in the construction sector (code 45 of the Italian classification of economic activities ATECO 1991) [26]. Descriptive analyses included the following variables: gender, age class (≤44 years old at diagnosis, 45–64, 65–84, >85); periods of diagnosis (1993–98, 1999–2003, 2004–08, 2009–13; 2014–18), anatomical sites (pleura, peritoneum, pericardium, tunica vaginalis of testis), level of diagnostic certainty (certain, probable, possible) and histopathological characteristics-limited to cases with histologically confirmed diagnosis (epithelioid, biphasic, sarcomatoid, not otherwise specified).

Mean age at diagnosis and year of first exposure, duration of exposure (considered as time-period from year of starting exposure to year of ending exposure) and latency (considered as time-period from year of starting exposure to year of MM diagnosis) have been calculated for construction workers and for cases exposed in occupational sectors other than construction.

Trends of asbestos consumption (tons), of all construction workers and of MM cases among construction workers (for 10-million-person years of observations) have been compared. Data about import, export and production of asbestos in Italy were obtained from the reports published by the Italian Institute of Statistics (ISTAT), the Italian Foreign Trade Institute, and the General Direction of Mines, Italian Ministry of Industry, as described elsewhere [27]. Asbestos consumption (defined as the domestic production plus the difference between imports and exports) between the end of Second World War and the asbestos ban in 1992 and the workforce in the construction sector from 1970 to 2015 (estimated combining historic data provided by ISTAT since 2010 and current figures [28]) were tabulated with the frequency of MM cases in construction workers (per 10 million of person/years) in the 1993–2015 period.

Furthermore, the distribution of cases exclusively exposed to asbestos in the construction sector was provided, with a detailed description of the economic sections in which they were employed.

The percentages of MM cases in construction workers (with respect to cases with a definite exposure) and other groups with different modalities of exposure (occupational other than construction, not occupational and unknown) have been calculated for periods of diagnosis.

Finally, the geographical distribution of crude incidence rates of mesothelioma in construction workers has been mapped, according to the municipality of residence of patient at diagnosis. For each municipality the population at denominator is the sum of person-years of observation which is not uniform among regions in the observed period (1993–2018).

For all statistical analyses, we used IBM-SPSS, version 25.0 (IBM SPSS, Armonk, New York, NY, USA) packages for statistical analysis and MapInfo Professional, version 17 (MapInfo Pro, Pitney Bowes, Stamford CT 06926-0700, USA).

## 3. Results

The ReNaM has collected 31,572 cases with diagnosis of MM in the period between 1993 and 2018.

The exposure evaluation was available for 24,864 MM cases (78.8%): among them 17,191 (69.1%) had an occupational exposure to asbestos and 3574 out of these (20.8%) had worked in the construction sector (Table 1).

The most of construction workers were males (99.4%), mainly in the age-class 65–84 (68.4%). The pleural site was prevalent (95.5%), the level of diagnostic certainty was high (87.2%) and the epithelioid subtype the most frequent (67.4%). Median duration of exposure in construction workers was longer with respect to the other occupationally exposed cases (31.0 years vs. 24.0), while figures of mean age at diagnosis, mean age, and year at first exposure were similar in the two groups, as well as the median latency (Table 2).

A clear decreasing trend in the construction sector workforce opposite to a regular increasing one in MM cases from construction sector was evidenced starting from 2008, while the two groups had similar growing tendencies in the previous years (Figure 1).

Moreover, together with the declining of asbestos consumption from the eighties to the ban in 1992, a decrease of construction workers is observed in the same period. The proportion of cases by modalities of exposure shows an increase of MM cases exposed in the construction sector from 11.1% in the period 1993–98 to 16.4% in 2014–2018 (Figure 2).

An exposure exclusively in the construction sector involved 2310 cases and concerned subgroups of economic categories that are reported in Table 3. Some carried out even more than one task and this results in 2454 exposures. The section of non-specific construction of buildings shows the highest number of exposures (65%), represented by the ‘general construction of buildings and civil engineering work’ (*n* = 361), ‘building of complete constructions or parts thereof; civil engineering’ (*n* = 251), and ‘construction’ without any specification (*n* = 983). A relevant number of exposures regard also the ‘plumbing’ (*n* = 283) and ‘installation of electrical wiring and fittings’ (*n* = 114) sections.

Finally, the geographical distribution of crude MM rates in the construction sector evidences the ubiquitous distribution over the national territory, with a range from 0.02 to 22.73 (cases per 100,000) (Figure 3).

## 4. Discussion

Although in the recent years the occupational trend in Italy evidenced a clear decrease in workforce, construction is still a relevant economic sector in Italy, accounting for 6.1% of all employees in 2015 and 24.6% of the whole industrial sector. Asbestos has been widely used in construction in many countries and MM cases with ascertained exposure during construction activities have been documented in various countries, such as South Korea [29], Netherlands [30], Germany [31], and also in USA [32], Sweden [33] and United Kingdom [19]. Until the ban in 1992, in Italy asbestos was intensively used in a large spectrum of applications and construction was one of the most involved economic sectors. The potential residual presence in buildings highlights the risk of exposure during activities dealing with or removing asbestos. Particularly, a sector at high risk is ‘other construction work involving special trades’, including the majority of asbestos abatement firms [34].

A key role is represented by the awareness of safety and correct perception of asbestos risk, from both companies and construction workers. There are critical points in this regard including the enormous diffusion of ACMs in buildings up to 1992 which are still in situ, and the lack of knowledge and compliance with current legislation. Most of the MM cases exclusively exposed in construction were employed in buildings constructing, where the widespread interchangeability of tasks between workers, especially in small and medium-sized construction companies, makes it difficult to recognize different profiles of risk. Construction activities were usually performed without using personal protective equipment for the respiratory tract and a study carried out in the Veneto Region evidenced a high lung content of asbestos fibers in various tasks [17]. An increasing number of studies demonstrated that the perception of occupational risks is influenced by a set of individual, social, cultural, and organizational variables [35,36,37]. Although it is common knowledge that asbestos is dangerous, many workers may be unaware of the actual degree of danger and the inherent risk situations [38]. In some studies on past asbestos exposed workers, an under-estimation of the intensity of personal asbestos exposure was described [39].

This minimization of exposure may correspond to a protective cognitive reaction to chronic occupational exposure, a form of risk denial. This cognitive mechanism can be overcome in sites where the community has developed a collective consciousness related to asbestos, such as Casale Monferrato [40].

In this context, the epidemiological surveillance of MM could increase the awareness of risk in workers, companies, stakeholders, and policy makers, providing evidence of the modalities of exposure to asbestos in environment and workplaces. Mesothelioma is recognized as an occupational disease. In Italy the public insurance system is managed by the Italian national workers’ compensation authority (INAIL) that, upon verification of the occupational origin of the disease, provides compensation for the received claims and benefits for to the subject or relatives.

This study confirmed the risk of asbestos exposure in MM cases among construction workers in Italy, already observed in past ReNaM Reports with reference to previous years of observation [41]. Almost 30 years after the asbestos ban, more than 1/5 of MM cases with a definite occupational exposure registered by ReNaM has suffered a causal exposure to asbestos in this sector and this percentage is significantly increasing in time. The geographical distribution of crude rates of MM in construction sector by municipalities symbolizes the industrial use of asbestos in this economic sector. Anyway, the municipality of residence at the time of diagnosis is taken as a proxy for exposure place, and this could be a possible source of bias when reading geographical maps.

The main strength of this study is related to the presence in Italy of a systematic active search of MM over the whole national territory, with standard criteria for active cases search, diagnosis classification, and evaluation of the occupational exposure to asbestos, obtained by means of a structured individual questionnaire. The temporal and spatial extent of the Italian surveillance system includes more than 1000 million of person-years of observations (from 1993 and for a large part of Italy). Mesothelioma epidemiological surveillance systems, comparable to the Italian experience for information completeness, exposure assessment and territorial coverage, are rare and (to the best of our knowledge) currently present only in Australia, France and South Korea [29,42,43,44]. At the same time, some limitations have to be taken account for the ReNaM network. The CORs activity did not begin at the same time and this could have introduced some bias in our study, given the not homogeneous territorial distribution of the construction sector. The comparison of trends reported in Figure 1 evidences the opposed tendencies between workforce and MM cases in the construction sector in the last years. Anyway, it must be interpreted with caution, taking account of the median 40 years of latency of mesothelioma. Moreover, asbestos consumption also includes economic sectors other than construction. Lastly, the systematic collection of cases started in the 1990 s only in some regions, in the residual only from 2000, allowing for a partial description of MM cases in Italy after asbestos exposures occurred during the last century. The exposure assessment is qualitative, and the ability to identify the specific modalities is not fully consistent among regional registries, with a percentage of collected exposure histories varying between 45% and 95% among regions. Furthermore, as generally is the case with specialized registries, potential over-reporting could be a concern for ReNaM [45].

## 5. Conclusions

In conclusion, our study provides evidence of a substantial and increasing fraction of MM cases in construction workers. There is a need to implement education and training for workers involved in activities such as remedial, maintenance, and building renovations, especially with reference to old buildings. Nevertheless, information about compensation claim is not always well-known among workers, especially when the economic sector of exposure is not previously recognized as “at risk”, and the level of compensation claim for mesothelioma is still lower than expected [46,47,48]. In this context, the role of health and safety institutions in increasing awareness and consciousness about the rights of compensation benefits should warrant equal opportunities for all of the exposed workers and the efficiency of insurance. The systematic surveillance of MM cases is a precious tool for identifying and control the potential and unknown sources of exposure.

## Figures and Tables

**Figure 1 ijerph-19-00235-f001:**
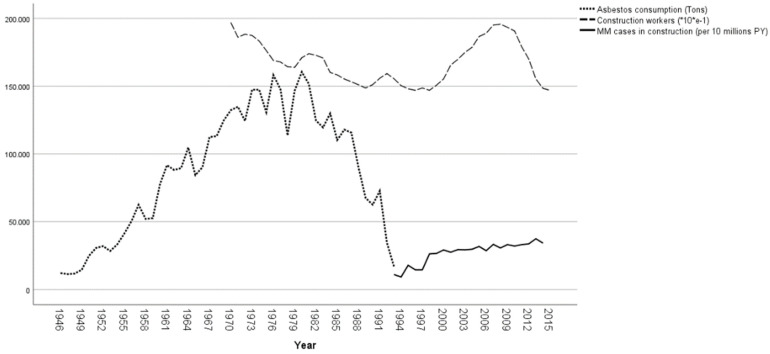
Asbestos consumption, construction workers and MM cases in construction workers (for 10-million-person years of observations). Italy, ReNaM, 1993–2015.

**Figure 2 ijerph-19-00235-f002:**
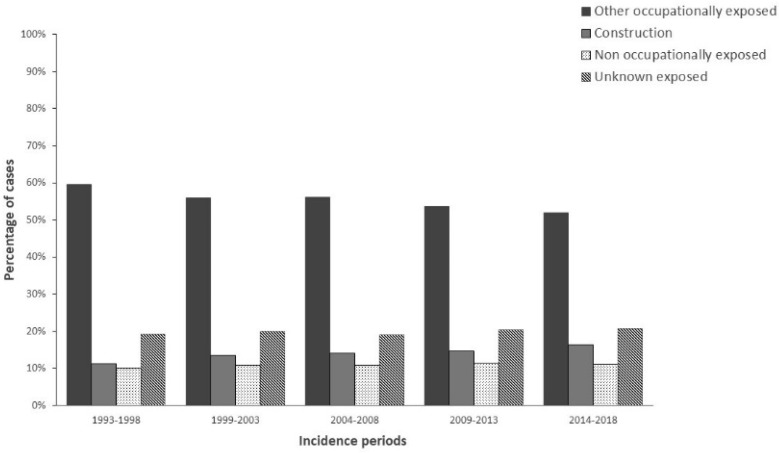
Distribution of MM cases (%) by exposed groups and time-period. Italy, ReNaM, 1993–2018.

**Figure 3 ijerph-19-00235-f003:**
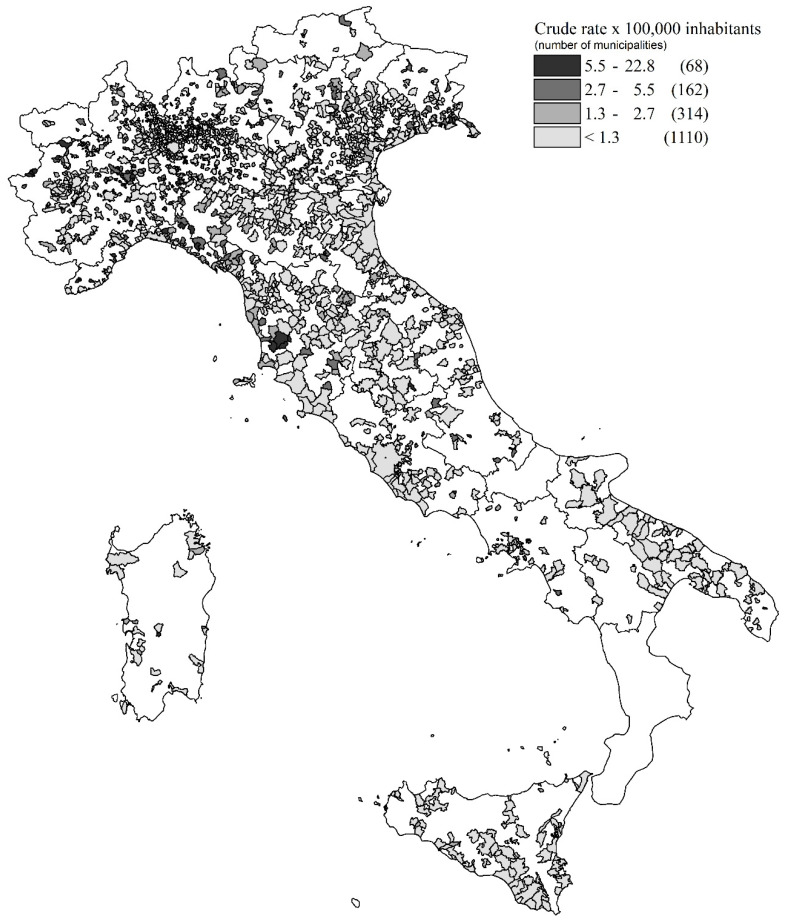
Map of crude rates of MM in construction sector by municipalities of patient’s residence. Italy, ReNaM, 1993–2018.

**Table 1 ijerph-19-00235-t001:** Distribution of MM cases (overall, occupational exposed and in the construction sector) by gender, age classes, incidence period, anatomical site, diagnostic certainty, histopathological characteristics, and modality of exposure. Italy. 1993–2018.

		All with Exposure Evaluation	%	Occupationally Exposed	%	Construction Workers	%
Gender	Men	18,256	73.4	14,959	87.0	3554	99.4
	Women	6608	26.6	2232	13.0	20	0.6
Age-classes	≤44	389	1.6	159	0.9	41	1.1
	45–64	6295	25.3	4404	25.6	936	26.2
	65–84	16,575	66.7	11,666	67.9	2446	68.4
	85+	1605	6.5	962	5.6	151	4.2
Incidence period	1993–1998	2010	8.1	1420	8.3	224	6.3
	1999–2003	4829	19.4	3348	19.5	651	18.2
	2004–2008	6034	24.3	4235	24.6	847	23.7
	2009–2013	6487	26.1	4428	25.8	952	26.6
	2014–2018	5504	22.1	3760	21.9	900	25.2
Anatomical site	Pleura	23,295	93.7	16,264	94.6	3413	95.5
	Peritoneum	1443	5.8	849	4.9	138	3.9
	Pericardium	49	0.2	28	0.2	8	0.2
	Tunica vaginalis testis	77	0.3	50	0.3	15	0.4
Diagnostic certainty	Certain	20,926	84.2	14,671	85.3	3116	87.2
	Probable	2050	8.2	1359	7.9	264	7.4
	Possible	1888	7.6	1161	6.8	193	5.4
Histopathological characteristics (cases with histologically confirmed diagnosis only)	Epithelioid	14,489	69.2	9996	68.1	2101	67.4
	Biphasic	2683	12.8	1968	13.4	397	12.7
	Sarcomatoid	1794	8.6	1358	9.3	303	9.7
	MM (NOS *)	1960	9.4	1349	9.2	315	10.1
Exposure modalities	Occupational	17,191	69.1				
	Familial	1278	5.1				
	Environmental	1067	4.3				
	Leisure-related	373	1.5				
	Unlikely	4955	19.9				
Total		24,864		17,191		3574	

* NOS, not otherwise specified.

**Table 2 ijerph-19-00235-t002:** Mean age at diagnosis, mean age and year of first exposure, duration of exposure and latency of MM by construction workers and other occupationally exposed subjects. Italy, ReNaM, 1993–2018.

	Construction Workers (*n* = 3574)	Other Occupationally Exposed Cases (*n* = 13,617)
Age at diagnosis (mean, SD *)	69.7 (±9.7)	70.3 (±9.7)
Age at first exposure (mean, SD *)	20.9 (±9.5)	22.5 (±7.9)
Year of first exposure (median, I-III quartile)	1959 (1952–1965)	1960 (1952–1967)
Duration of exposure (median, I-III quartile)	31.0 (16.0–40.0)	24.0 (10.0–35.0)
Latency (median, I-III quartile)	50.0 (42.0–57.0)	49.0 (41.0–56.0)

* SD, standard deviation.

**Table 3 ijerph-19-00235-t003:** Occurrences of exposures by economic activity (subjects exclusively exposed in the construction sector and carrying out even more than one task.

Code.	Description of Activity	N	%
45.00.0	Construction	983	40.1
45.21.0	General construction of buildings and civil engineering work	361	14.7
45.33.0	Plumbing	283	11.5
45.20.0	Building of complete constructions or parts thereof; civil engineering	251	10.2
45.31.0	Installation of electrical wiring and fittings	114	4.6
45.32.0	Insulation work activities	87	3.5
45.45.0	Other building completion	79	3.2
45.40.0	Building completion	45	1.8
45.22.0	Erection of roof covering and frames	36	1.5
45.23.0	Construction of motorways, roads, airfields and sport facilities	33	1.3
45.24.0	Construction of water projects	33	1.3
45.43.0	Floor and wall covering	27	1.1
45.25.0	Other construction work involving special trades	26	1.1
45.11.0	Demolition and wrecking of buildings; earth moving	23	0.9
45.41.0	Plastering	21	0.9
45.10.0	Site preparation	18	0.7
45.34.0	Other building installation	12	0.5
45.30.0	Building installation	8	0.3
45.44.0	Painting and glazing	8	0.3
45.12.0	Test drilling and boring	5	0.2
45.42.0	Joinery installation	1	0.0
Total		2454	100.0

## Data Availability

Data sharing not applicable (restrictions apply to the availability of these data).

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
