# Peer review of "Asbestos Exposure and Malignant Mesothelioma in Construction Workers—Epidemiological Remarks by the Italian National Mesothelioma Registry (ReNaM)"

_ijerph, 2021, doi:10.3390/ijerph19010235_

Round 1
Reviewer 1 Report
In the present manuscript” Asbestos exposure and malignant mesothelioma in construction workers. Epidemiological remarks by the Italian national mesothelioma registry (ReNaM)” Binazzi et al comprehensively and clearly describe the interesting rise of MM incidence in construction workers by using a national registry. This is a well written and important manuscript from my point of view.
Only one minor issue comes up from my side: can you give more information about compensation in Italy and who is reporting to the registry mainly?
Are the main reporting people pathologists after pathological diagnosis or physicians?
Who pays the compensation?
Is it the patient only who will get compensation or also his/her family?
Is it the insurance or the state who pays? Please give some more details about this.
Author Response
Only one minor issue comes up from my side: can you give more information about compensation in Italy and who is reporting to the registry mainly?
Answer: a public insurance system is active in Italy, leaded by the Italian national workers’ compensation authority (INAIL), covering about 80 % of the Italian workforce. INAIL receives claims for occupational diseases from all national territory and provides compensation, after verifying the occupational origin of the disease. A sentence about it has been included in the text.
Are the main reporting people pathologists after pathological diagnosis or physicians?
Answer: for every request for compensation, INAIL physicians visit the sick subject, verify the existence of causal relationship with the working activities according to forensic criteria and assign economic and healthcare benefits.
Who pays the compensation?
Answer: The INAIL allows compensation in response to individual workers’ claims, and benefits are granted to the subject or to relatives.
Is it the patient only who will get compensation or also his/her family?
Answer: Mesothelioma is recognized as an occupational disease, but environmental (living in the neighborhood of an industrial or natural source of asbestos) or indirect exposures (living with a person occupationally exposed to asbestos) have also been associated with it. The INAIL fund for asbestos victims provides compensation also for environmental or in-house exposures, or heirs.
Is it the insurance or the state who pays? Please give some more details about this
Answer: See above
Reviewer 2 Report
General comments
This is an interesting, reliable and well-structured article providing clear and convincing evidence of occupational exposure to asbestos being a main risk factor for the development of malignant mesothelioma. It is certainly worth to be published even though it, basically, is a confirmation of previous knowledge or at least of previous assumptions. In the specific comments, I have compiled a number of proposals for amendments and clarifications that might help to further improve the quality of the manuscript.
Specific comments
Abstract: On line 67, I would replace "asbestos consumption" by "asbestos exposure". Consumption is usually related to ingestion. Or do you mean the total use of asbestos in Italy rather than individual exposure? (I have got this impression later from the text. However, the abstract should be clear about the meaning.)
Introduction: Lines 82/83 - are these "per year" incidences? Also, it might be useful to give, just for comparison, one or two additional figures for other highly developed countries such as Italy to put the 1:100,000 incidence into perspective. A "world" incidence might be an inapropriate figure for comparison if it is the only one given.
Lines 114/115 - I assume that the unit f/cc means fibres in cubic centimeter. If this is not true, please specify the unit because it might be misleading.
Line 116 - Should be perhaps better "... to analyse the proportion of constructions workers among all mesothelioma cases ...". Not the workers are subject to analysis but the number of affected persons. Also, I would suggest, for better readibility and understanding, to split the single sentence of which the last paragraph consists into two. The aim of the study and the points of discussion should be separated.
Materials and methods: Lines 126 - 131 - Even though its meaning is completely clear, this sentence might be a bit difficult to swallow. My proposal would be: With regard to diagnosis of MM, the cases were allocated to one of three classes that are distinguished by an increasing level of uncertainty: "certain" (...), "probable" (...), and "possible" (...).
Line 133 - I wonder what "familial" exposure might be. In-house exposure?
Lines 153/154: "Certain level of diagnosis" seems not very clear. I guess that "cases with histologically confirmed diagnosis" are meant here. If so, "morphology" could be replaced and specified by "histopathological characteristics". But then, "biphasic" is surprising because this is not a usual pathological term. Could you, please, clarify and explain in more detail what that diagnosis means?
Line 155 - Should be "mean age at and year of first exposure". ("Year at" sounds a bit odd.)
Results: Lines 187/188 - Of course, I don't doubt the figures but I have my doubts if they are in line with the 1:100,00 incidence as reported for Italy before. Assuming a population of 60 millions in Italy, one would expect 600 cases of MM per year. For the 26-year interval considered, this would result in an estimated number of cases in the magnitude of 15,600, i.e., ca one half of the reported incidence. Is there any explanation for this discrepancy? What is wrong?
Table 1 - In the caption, the term "overall" might be misleading and could be replaced, perhaps, by "all with exposure evaluation".
Line 219 - What is "exposure circumstances by economic activity". I simply don't understand it and I presume that other readers likewise will not.
Figure 3 - Indeed, the distribution of MM cases is ubiquitous but remarkably unequal. Is that only due to the reporting deficiencies from some regions such as Sardinia or Calabria or are there other reasons?
Discussion: Line 244 - Should be sectors, shouldn't it? (Instead of "sector".)
Line 254 - Should be "makes it difficult".
Lines 261 - 265 - Self-underestimation of exposure does not mean reduction of exposure but only a decreased perception of risk. And what exactly is the "protective cognitive reaction to chronic occupational exposure"? I have an idea what it might be but some more explanation would be useful.
However, you might wish to check the quality of Figure 1. It took me some time to recognize that there are indeed three separate curves.
Author Response
Specific comments
Abstract: On line 67, I would replace "asbestos consumption" by "asbestos exposure". Consumption is usually related to ingestion. Or do you mean the total use of asbestos in Italy rather than individual exposure? (I have got this impression later from the text. However, the abstract should be clear about the meaning.)
Answer: the word ‘consumption’ has been replaced by ‘use’
Introduction: Lines 82/83 - are these "per year" incidences? Also, it might be useful to give, just for comparison, one or two additional figures for other highly developed countries such as Italy to put the 1:100,000 incidence into perspective. A "world" incidence might be an inapropriate figure for comparison if it is the only one given.
Answer: incidence data are referred to 2020. Other figures for highly developed countries have been added
Lines 114/115 - I assume that the unit f/cc means fibres in cubic centimeter. If this is not true, please specify the unit because it might be misleading.
Answer: f/cc stands for Fibers per Cubic Centimeter
Line 116 - Should be perhaps better "... to analyse the proportion of constructions workers among all mesothelioma cases ...". Not the workers are subject to analysis but the number of affected persons. Also, I would suggest, for better readibility and understanding, to split the single sentence of which the last paragraph consists into two. The aim of the study and the points of discussion should be separated.
Answer: the sentence has been modified and split into two
Materials and methods: Lines 126 - 131 - Even though its meaning is completely clear, this sentence might be a bit difficult to swallow. My proposal would be: With regard to diagnosis of MM, the cases were allocated to one of three classes that are distinguished by an increasing level of uncertainty: "certain" (...), "probable" (...), and "possible" (...).
Answer: the sentence has been modified accordingly
Line 133 - I wonder what "familial" exposure might be. In-house exposure?
Answer: the term ‘familiar’ has been replaced by ‘in-house’
Lines 153/154: "Certain level of diagnosis" seems not very clear. I guess that "cases with histologically confirmed diagnosis" are meant here. If so, "morphology" could be replaced and specified by "histopathological characteristics". But then, "biphasic" is surprising because this is not a usual pathological term. Could you, please, clarify and explain in more detail what that diagnosis means?
Answer: the sentence has been modified accordingly. Malignant biphasic tumors contain a combination of epithelioid and sarcomatoid cells
Line 155 - Should be "mean age at and year of first exposure". ("Year at" sounds a bit odd.)
Answer: the terms have been modified accordingly.
Results: Lines 187/188 - Of course, I don't doubt the figures but I have my doubts if they are in line with the 1:100,00 incidence as reported for Italy before. Assuming a population of 60 millions in Italy, one would expect 600 cases of MM per year. For the 26-year interval considered, this would result in an estimated number of cases in the magnitude of 15,600, i.e., ca one half of the reported incidence. Is there any explanation for this discrepancy? What is wrong?
Answer: Descriptive statistics reported in the ‘Results’ section concern the entire ReNaM archive, that includes MM cases diagnosed in the period 1993-2018. With regard to incidence data, the collection is partial in three regions (Molise, Calabria and Sardinia) and incident case-list is still incomplete. Moreover, it must be considered that the temporal development of the ReNaM dataset has been not homogenous: some regions started collecting incidence cases even before 1993, others started later or partially contribute, making any evaluation of the MM incidence trend strongly limited. These issues are reported in the discussion.
Table 1 - In the caption, the term "overall" might be misleading and could be replaced, perhaps, by "all with exposure evaluation".
Answer: the term has been modified accordingly
Line 219 - What is "exposure circumstances by economic activity". I simply don't understand it and I presume that other readers likewise will not.
Answer: the sentence has been modified accordingly.
Figure 3 - Indeed, the distribution of MM cases is ubiquitous but remarkably unequal. Is that only due to the reporting deficiencies from some regions such as Sardinia or Calabria or are there other reasons?
Answer: The geographical distribution of crude rates of MM in construction sector by municipalities may be considered a picture of the industrial use of asbestos in this economic sector. However, when reading territorial maps, a possible source of bias is due to the municipality of residence at the time of diagnosis taken as a proxy for exposure place. Also the heterogeneous spatial and temporal reporting from Italian regions is to be considered in the geographical mapping of MM cases. These issues have been added in the text.
Discussion: Line 244 - Should be sectors, shouldn't it? (Instead of "sector".)
Answer: the term has been corrected
Line 254 - Should be "makes it difficult".
Answer: the sentence has been modified accordingly.
Lines 261 - 265 - Self-underestimation of exposure does not mean reduction of exposure but only a decreased perception of risk. And what exactly is the "protective cognitive reaction to chronic occupational exposure"? I have an idea what it might be but some more explanation would be useful.
Answer: Self-underestimation of exposure does not mean reduction of exposure but a decreased perception of risk that can also be a form of denial. The underestimation of risk is a defense mechanism on the psychic level. Defense mechanisms are protective cognitive reactions to lower anxiety levels
However, you might wish to check the quality of Figure 1. It took me some time to recognize that there are indeed three separate curves.
Answer: the quality of Figure 1 has been improved